# Pixel Image Analysis and Its Application with an Alcohol-Based Liquid Scintillator for Particle Therapy

**DOI:** 10.3390/s22134876

**Published:** 2022-06-28

**Authors:** Ji-Won Choi, Ji-Young Choi, Hanil Jang, Kyung-Kwang Joo, Byoung-Chan Kim

**Affiliations:** 1Institute for Universe & Elementary Particles, Department of Physics, Chonnam National University, Yongbong-ro 77, Puk-gu, Gwangju 61186, Korea; 218396@jnu.ac.kr; 2Department of Fire Safety, Seoyeong University, Seogang-ro 1, Puk-gu, Gwangju 61268, Korea; hjang@seoyeong.ac.kr; 3Medical Radiation, Wonkwang Health Science University, 514, Iksan-daero, Iksan-si 54538, Korea

**Keywords:** liquid scintillator, alcohol, electron beam, percent depth dose, phantom, linear accelerator, novalis Tx, CT, digital camera, image analysis, medical physics, Radon transform, optical tomography

## Abstract

We synthesized an alcohol-based liquid scintillator (AbLS), and we implemented an auxiliary monitoring system with short calibration intervals using AbLS for particle therapy. The commercial liquid scintillator used in previous studies did not allow the user to control the chemical ratio and its composition. In our study, the chemical ratio of AbLS was freely controlled by simultaneously mixing water and alcohol. To make an equivalent substance to the human body, 2-ethoxyethanol was used. There was no significant difference between AbLS and water in areal density. As an application of AbLS, the range was measured with AbLS using an electron beam in an image analysis that combined AbLS and a digital phone camera. Given a range–energy relationship for the electron expressed as areal density, the electron beam range (cm) in water can be easily estimated. To date, no literature report for the direct comparison of a pixel image analysis and Monte Carlo (MC) simulation has been published. Furthermore, optical tomography of the inverse problem was performed with AbLS and a mobile phone camera. Analyses of optical tomography images provide deeper insight into Radon transformation. In addition, the human phantom, which is difficult to compose with semiconductor diodes, was easily implemented as an image acquisition and analysis system.

## 1. Introduction

As one of the standard dosimeters, a water phantom composed of water, an acrylic container, a semiconductor diode, etc., has been used for quality support operations for a linear accelerator (LINAC). Quality calibrations with water phantoms are performed at the initial LINAC installation stage, and then conducted both quarterly and annually. [1,2]. However, water phantoms are time-consuming, and there is a risk of electric shock. To overcome the disadvantage of water phantoms, it has been proposed to use tissue-equivalent ion chambers, optical fibers, and plastic scintillators instead of water [3,4]. For short calibration cycles of quality support, the usage of fiber-coupled scintillators or bulk-up tissue-equivalent ion chambers was suggested. Meanwhile, in existing medical physics research, fluorescence images are acquired and analyzed using a scintillator combined with various devices. However, to date, there are no fluorescence image analysis results that are comparable to the Monte Carlo (MC) result [5,6,7,8]. In this paper, for fluorescence image analysis, two-dimensional percent depth dose (2D PDD) results using a liquid scintillator (LS) were acquired and directly compared with those of MC simulation.

Furthermore, as an application, a sinogram analysis was performed with a simple experimental configuration, using AbLS, a rotating plate, and a mobile phone camera. Tomographic images were reconstructed by inverse Radon transformation of the sinogram. We hope that this application will provide a suitable choice for a variety of medical physics and simple clinical settings for particle therapy purposes.

## 2. Introduction

### 2.1. Brief Synthesis of AbLS

In a typical planar tissue-equivalent ion chamber, it is difficult to measure the beam profile for high-energy electron beams exceeding the thickness of the plate due to the limited effective ionization volume [9,10]. In addition, since the plastic scintillator is vulnerable to radiation damage, defects easily occur near the color center. Furthermore, it is problematic that the plastic scintillator is easily discolored due to a polymerization reaction [11,12].

In previous studies, the LSs used for phantoms were all commercial LSs [13,14]. In those cases, it is difficult for users to check information other than the specification provided by the manufacturer. In this study, an AbLS was synthesized using a single solvent that is capable of simultaneously mixing water and alcohol. The effective atomic number and soft tissue density are similar to those of the human body. AbLS, which uses alcohol as a solvent, is an attempt not seen anywhere else. For the alcohol, 2-ethoxyethanol with balanced lipophilicity and hydrophilicity was used, since it has a higher density than alcohol. Therefore, one advantage is that the user can easily select the synthesis procedure and the chemical composition ratio for the LS. Organic, solvent-based LSs in high-energy physics used an optimized primary fluor of ~3 g/L anda secondary fluor composed of ~30 mg/L [15,16]. In our study, 2,5-diphenyloxazole (C_15_H_11_NO, PPO) was used as the primary fluor, and 1, 4-bis(2-methylstyryl) benzene (C_24_H_22_, bis-MSB) was added as the secondary fluor. When synthesizing an AbLS, the same fluor concentration used in the reactor neutrino experiments is maintained [17,18]. PPO and bis-MSB offer relatively good solubility in 2-ethoxyethanol. After dissolving, ultrapure water was added to dilute the solution until it was completely mixed using a magnetic bar. The optimized ratio of 2-ethoxyethanaol to water was 7:3. Optical and physical properties of the AbLS were described in detail elsewhere [15,16].

### 2.2. Practical Range, Absorbed Dose, and Beam Profile in Electron Beam Therapy

The success or failure of particle therapy depends on how accurately the range and dose are predicted and how accurately they are measured. The beam path length is commonly referred to as the range of a charged particle. Here, range means the distance where the particle stops. The analytical form or continuous slow-down approximation range (CSDA range) offers a good representation of the average value of the beam path length [19,20]. The CSDA range is a physical quantity used to shield charged particles that include alpha, beta, proton, heavy ions, etc. The particle beam used in particle therapy delivers energy to the target tissue by penetrating the body surface. Deoxyribose nucleic acid (DNA) chain damage or apoptosis, which are the main consequences of physicochemical and molecular biological effects caused by energy transfer effects in cell constituents, are predominant [21]. Since these phenomena are induced by the incident radiation, the depth at which the particles penetrate into tissue is important. This penetration depth is called the projected range or depth. This projected range is determined by the PDD and is usually not equal to the beam path length. Since the definition of the projection range depends on the intended usage, in this study, the practical range (R_p_) for particle treatment and shielding was used as the range. R_p_ is commonly defined as the depth corresponding to the intersection between the tangent of the curve rapidly decreasing at the point of maximum dose on the PDD plot and the horizontal line passing through the lowest point contributed by bremsstrahlung [22].

The dose is the energy absorbed by the medium per unit mass, and it is proportional to the stopping power of the incident-charged particles. The dose has various definitions, because of the difference between the energy losses of a particle and the energy absorbed by the medium, that take into account various factors, such as binding energy, excitation energy, and heat dissipation. In particle therapy and shielding, an effective dose is used. However, in this study, an absorbed dose was used to maintain consistency and simplicity. From the particle therapy point of view, the dominant factor between range and dose is range. Determining the maximum therapeutic dose and the number of beam exposure splits is based on different recovery cycles between the target cancer tissues and normal tissues [23].

Geometric and dosimetric beam field sizes differ due to the interaction between the particle beam and the material. The LINAC dosimetric beam profile can only be known by measuring it with a dosimeter. To reproduce this circumstance, beam factors such as width, homogeneity, and cross-sectional shape were used as major input values for MC simulation [24].

### 2.3. Colorimetry for a Fluorescent Image of AbLS

Composite metal-oxide semiconductor (CMOS) image sensors apply red, green, and blue (RGB) filters to mimic the response of human cells to visible light. In 1931, the International Commission on Lighting (CIE) standardized and quantified the visual response from a practical and modern perspective [25]. The CIE proposed the concepts of color-matching functions, chromaticity diagrams, and color space. These concepts became the basis of modern colorimetry. Color space is also used to encode or decode an image signal as it is converted by a digital camera into an image [26].

Generally, RGB is used to express luminance and color. However, there is a problem in that color and brightness information cannot be expressed or separated, respectively [27]. This means that color space is required to represent each piece of information, that is, a standard orthogonalization is required to represent the color and brightness information. Similar to the XYZ color space proposed by the CIE in human visual response experiments, nonlinear transformations from RGB to HSV and HSV color spaces have been proposed to solve the above problems [28]. HSV stands for hue (H), saturation (S), and value (V). An advantage of HSV space is that brightness and color can be represented and separated by one H and one V reference, respectively. Utilizing the HSV space for image processing can save a lot of computational time and resources.

### 2.4. Sinogram as an Application of Inverse Problems

From a more practical point of view, inverse problems can be considered a parametric estimation of statistical inference [29,30]. Computed tomography (CT) is one typical example of a classical application of the inverse problem. The main physical parts of data acquisition using CT are the target, the beam source, and the detector. The target is at the center of a straight line connecting the beam source and the detector. The beam generator and the detector rotate while facing each other. While rotating, the detector measures the attenuated intensity as the penetrated beam from the generator to the target. The physical properties of soft tissue considered in medical physics are the effective atomic numbers 6–7 and densities of about 1 g/cm^3^ [31]. The average photon energy generated in the CT peak voltage of 120–140 kV_p_ region is approximately 70 keV [32,33]. Given the effective atomic number, physical density, volume, and photon energy range of a material of interest in CT, the significant parts of photon interaction with matter are the photoelectric absorption and Compton scattering. Photoelectric absorption dominates below a few kiloelectronvolts, and Compton scattering dominates above a few kiloelectronvolts. Furthermore, multiple-scattered photons are more absorbed within the target volume. This point becomes more significant in optical tomography. The pixel array with the intensity data obtained during the CT acquisition process is called a sinogram [34]. If the inverse Radon transformation is performed on the sinogram, a tomographic image of the object can be reconstructed.

## 3. Experiment and MC Simulation

The configuration of the therapeutic LINAC, AbLS sample, and digital camera is shown in Figure 1. The distance between the head of the LINAC and the AbLS sample was 1 m, and the distance between the AbLS sample and the camera was 50 cm. When 6, 9, and 12 MeV electron beams were irradiated, a 10 cm length × 5 cm width × 10 cm height rectangular acrylic container was used. When irradiating with gamma rays, the same rectangular acrylic container was used, except for the width of 20 cm. The top of the container where the beam strikes directly was open. The cross-sectional size of the electron beam was 6 × 6 cm, and that of the gamma ray was 20 × 20 cm. Both containers were filled with AbLS, and the beams were irradiated perpendicular to the open side of the container to capture the fluorescence emitted from the AbLS sample. The reason for using gamma rays was to check if scintillating light was emitted for neutral particles. A Canon 450D was used to photograph the fluorescence emitted after the electron beam was incident on the AbLS sample. The shooting and video recording conditions were set by the night shooting mode recommended by the manufacturer, and the exposure time was set to 1 s. For cross-checking, a Samsung Galaxy 10 series mobile phone was used as the rear camera to film the fluorescent light emitted after the photon beam was incident on the AbLS sample. The distortion effect of the refractive index was minimized by making the axis of the front of the container and the camera lens perpendicular. JPG, MP4, and H.264 file formats were analyzed in “Windows Subsystem for Linux2 (WSL2)”. Image extraction from the video was performed using a C-based open-source package, and image analysis was conducted using a Python-based package for fast implementation and productivity [35,36].

Figure 2a–c represent the fluorescent light of AbLS using 6, 9, and 12 MeV electron beams, and Figure 2d shows the fluorescence of gamma rays originating from electrons accelerated to 6 MV. The fluors we dissolved in AbLS typically emit light of a blue wavelength. The higher the energy of the incident electron beam, the longer the penetration depth of the electron beam. In the case of gamma rays, their energy was too high, and as a result, they passed through the container. Therefore, the entire area of the container was colored blue. Each fluorescence picture has approximately 4000 × 3800 pixels. Decomposed histograms based on CIE XYZ for each fluorescence image are shown in Figure 2e–g. Lowercase letters (xyz) represent normalized distribution. RGB bases were mapped 1:1 to XYZ bases through matrix transformation, and the blue component was dominant. The histograms for the Z basis for the blue component in each electron energy showed a similar shape. According to the XYZ color-matching function, blue mainly contains a weak x component and a strong z component.

Reconstructed images of fluorescent light in HSV space with 6, 9, and 12 MeV electron beams and gamma rays are shown in Figure 3. As seen in Figure 2, the blue component was dominant in the original image. As a result of this, a lot of blue color in the single H-based decomposition was seen in Figure 3a–d. Figure 3e–h show a single S-basis achromaticity. Darker colors represent more achromatic colors, while lighter colors represent more colorful properties. Figure 3i–l show luminance levels represented by a decomposed single V basis. Instead of a grayscale representation, an arbitrary color was used to make them more visible.

A histogram of V values for each fluorescence image is shown in Figure 4a. Figure 4b,c were reconstructed images of a fluorescence image of a 12 MeV electron beam. Figure 4d–f show reconstructed images of a fluorescence image of a gamma ray. Figure 4b–f show differences in image reconstructions when V thresholds of 0.1, 0.3, 0.1, 0.3, and 0.6 were applied, respectively. As expected, the regions where the AbLS sample received a lot of energy from the incident particle beam had high V values in the reconstructed image. As the V value increased, the background around the central image was removed. For quantitative image pixel analysis, V-cut was used to distinguish between signal and noise in a histogram of V values.

With the gamma ray, a double peak was seen in the V distribution, as shown in Figure 4a. One peak near 0.3 of the distribution was caused by a reflected fluorescent light from the top and bottom of the container shown in Figure 2d. Another peak near 0.6 of the distribution was a direct emission light of AbLS. Increasing the V-cut value removed more noise and minimized signal loss.

As a first attempt to cross-validate the image pixel analysis, MC simulation was implemented for the interaction between AbLS and the particle beam [37,38]. Since the density of AbLS was 0.945 g/cm^3^ and the amount of fluor was very little, fluor was neglected when calculating the effective atomic number and average ionization energy. The effective atomic number of AbLS was 6.73, and the average ionization energy was about 68.8 eV [39]. It is difficult to track individual particles in general-purpose detectors. Consequently, beam path length cannot be measured, and only projected range can be measured. However, MC simulations can virtually track particle movement and interactions through sampling and labeling. The particle beam transport part and interaction in the MC simulation were implemented without using a phase-space database or voxelization [40,41]. The PDD received by AbLS from the particle beam was implemented using only the base classes in the MC simulation package, which provide a user-friendly code.

The dose distribution of the fluorescence imaging of Figure 5a–d and the MC of Figure 5e–h is shown in Figure 5. The color of the contour area indicates the amount of energy absorbed by AbLS. Going from purple to red means more energy absorption. The dose distributions of the fluorescence distributions of the reconstructed image and the MC were almost identical. The MC reproduced the pixel image reasonably well. Looking at the contour color between the reconstructed fluorescence image and the MC, the ionization energy of AbLS in the MC’s case was slightly lower than that of the reconstructed fluorescence image. The consistent dose contour distribution indicated that the same energy transfer mechanism worked well between the fluorescent images and the MC with AbLS. However, the discrepancy between the contours of the MC and the actual fluorescence images shows that sufficient energy was not delivered to the fluor to emit light. Figure 5a–c show pixels with a V value of 30% or more that were selected among the 6, 9, and 12 MeV electron beam fluorescence images, respectively. Figure 5d shows pixels with V values of 60% or more that were selected in the 6 MeV photon fluorescence image. The 2D PDD MC simulation results for AbLS with 6, 9, and 12 MeV electron beams and a gamma ray are shown in Figure 5e–h, respectively. The solid blue line indicates the boundary of the container, the dashed blue line indicates the open side of the container, and the white hatched area represents the area profile of the incident particle beam. As a result of the comparative analysis between the fluorescence image and the MC simulation, there was no actual light emission in the region receiving less energy than the excitation and ionization energies required for fluorescence emission. In an actual LINAC, the emitted electron beam density was high and uniform, but the simulated electron beam density was low and nonuniform. This made a difference between the reconstructed electron beam fluorescence image and the MC-simulated 2D PDD. The basic implementation of the particle beam showed sufficient dose characteristics of the electron beam. When more computational resources are available, a more accurate beam implementation can be obtained by using a phase-space database and voxelization technique.

As seen in Figure 5d,h, there were also differences between the reconstructed photon beam fluorescence images and the MC-simulated 2D PDD. This phenomenon occurred because the photon beam was generated in the form of bremsstrahlung in the LINAC. In a LINAC, a high-energy photon beam occurs when it collides with a target material such as tungsten. The flattened filter of the LINAC head must ensure that bremsstrahlung energy distribution is uniform for the direction of incidence of electrons.

Figure 6a,b show an experimental configuration of the AbLS and cell phone combination. A UV lamp was positioned approximately 10 cm above the AbLS sample with a T-shaped bar. A partition with a tiny hole was installed to prevent stray light from directly entering the camera. Only the desired light reached the camera. The emission wavelength of the UV lamp was 365 nm. A 3 cm diameter 100 mL vial filled with AbLS had a T-shaped bar with a microscopic hole in the center of the bar. The holes in the T-shaped bar immersed in the AbLS vial can be seen. The AbLS sample vial was placed on an electric rotary disk that revolved once every 50 s and videoed with the cell phone camera. The best 6000 JPG rotated images were extracted at 30 frames per second. The baseline for reconstructing the sinogram was the red solid line. The sinogram was reconstructed from z = 451 pixel lines of each image.

## 4. Results

### 4.1. Range Measurement

As shown in Figure 2d, it was difficult to measure the range of the gamma ray with the small acrylic container. Figure 7a shows that the calculated CSDA ranges for AbLS with 6, 9, and 12 MeV electron beams were 3.04, 4.50, and 5.89 g/cm^2^, respectively. Because the effective atomic numbers of water and AbLS were almost the same, the mean ionization potential energy of water was about 10 eV, and that of AbLS was about 68 eV [39,42]. Divided by the density of AbLS, the CSDA ranges were 3.22, 4.76, and 6.23 cm.

Figure 7b shows that the projected ranges with MC simulations were 3.11, 4.68, and 6.28 cm. Since there was refraction of the sample container material, distortion caused by the presence or absence of AbLS in the sample container was considered [43]. Taking this effect into account, the measured projected ranges of the 6, 9, and 12 MeV electron beams were 3.17 ± 0.30, 4.75 ± 0.30, and 6.41 ± 0.60 cm, respectively. All charged particles exhibit a Bragg peak in the 1D PDD. As seen in Figure 7b, not all MeV electrons reached maximum dose at the surface or skin, but they built up to a maximum dose. In 1D PDD, this phenomenon is called build-up, and this part is not dealt with in detail in medical physics and nuclear engineering; however, our MC fully reflected it. This was a result of an increase in the charged particle equilibrium area due to the secondary particle transport process. The main factor considered in the electron transport phenomenon was the multiple scattering of electrons [22,44,45]. This part is not emphasized in the medical physics or nuclear engineering fields.

### 4.2. Sinogram Analysis

Figure 8a shows the T-shaped bar placed on the CT examination table before scanning. Figure 8b–d represent reconstructed CT images of the transverse plane, median plane, and frontal plane, respectively, for the T-shaped bar. As shown in the figure, images with high resolution can be reconstructed by CT technology. To make a relative comparison with the CT image results, sinogram analysis through Radon transformation was performed with photo images obtained in Figure 6. Figure 9 shows the optical tomography image obtained by inverse Radon transformation of the sinogram. Figure 9a represents the 2D pixel array sinogram reconstructed from photo images. The tomographic top view image of Figure 9a is shown in Figure 9b,c, where white represents the bright part and black corresponds to the dark part. In Figure 9b, a narrow scale contrast was applied. The large outer circle shown here is the boundary of the glass bottle, while the small inner circle is the cross-sectional boundary of the T-bar. The two rectangles located at the coordinates (150, 100) and (150, 200) are cylinder-shaped holes across the center of the cross-section of the T-shaped bar. Due to the narrow contrast range, the center of the rectangle from (150, 110) to (150, 190) is the part that connects the upper rectangle and the lower rectangle and is not visible. In Figure 9c, a wide-scale contrast was applied. The upper and lower rectangular connection, which is not visible in Figure 9b, could be seen. Unlike CT images, which use X-rays, the pin-wheel-shaped image at the center was generated by optical effects, such as the reflection or refraction of visible light. If there were no such defects, it would appear in the form of a straight-line pattern. At the same time, a small circle appeared at the center of the cross-sectional T-shaped bar. These are artifacts created by ill-posed conditions, in which many solutions exist in inverse problems. Artifacts that occur when implementing inverse problems are largely influenced by both the detection geometry and physical situations that can affect the projection process in the Radon transformation process. In the absence of axial symmetry, double sampling (or 360-degree sinogram) may reduce these geometric artifacts more than when using a single sampling (or 180-degree sinogram) [46,47]. In addition, as shown in Figure 9c, physical artifacts were caused either by assumptions not holding, such as parallel beam attenuation or geometric optics, or by very severe diffraction and refraction. Comparing the image analysis taken with the mobile phone camera to the CT image, the resolution is lower, but it showed sufficient ability to identify the major features of the object.

## 5. Conclusions

By synthesizing AbLS, a quality assurance and quality control system with a short calibration cycle that aids standard dosimeters was implemented. Unlike previous studies using a commercial LS, our AbLS based on 2-ethoxyethanol has the advantage of being able to control the chemical composition and its ratio within a soluble range. The calculated specific electron CSDA range (g/cm^2^) and the simulated specific electron projected range (g/cm^2^) were not significantly different between AbLS and water. The specific range makes it easy to inversely calculate the range (cm) of water used for particle therapy in medical physics. A cross-check of the image pixel analysis was implemented and validated using MC simulations for the interaction between AbLS and incident particle beams. In the MC simulations, the electron beam ranges using AbLS were 3.11, 4.68, and 6.28 cm. In addition, in the image pixel analysis combining AbLS and the camera, the electron beam ranges of AbLS were measured to be 3.17, 4.75, and 6.41 cm.

As an application of inverse problems, optical tomography was implemented with a combination of AbLS and the mobile phone camera. Sinogram results were directly compared with CT imaging analysis images. Although the resolution was low, the technique showed sufficient ability to identify the major features of the object. Unlike analyses using the energy applied in conventional radiation and astronomy, analyses of optical tomography images in the visible region provide insight into the physical meaning of Radon transformation and artifacts. We expect these implementations with AbLS will be used as human phantoms for the complex lung and head, which are difficult to construct with semiconductor diodes.

## Figures and Tables

**Figure 1 sensors-22-04876-f001:**
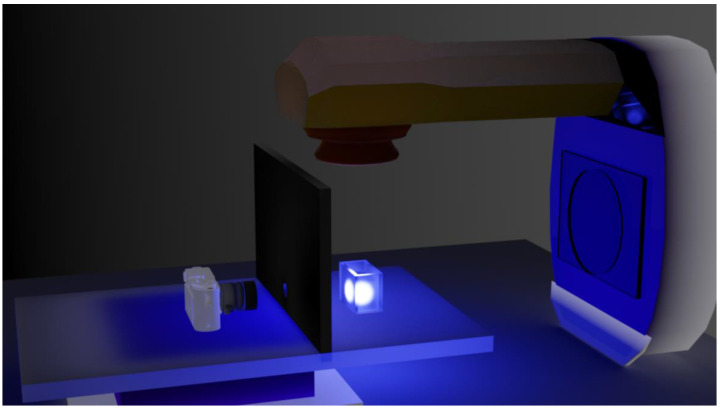
Schematic of the configuration of LINAC, AbLS sample, and digital camera. The distance between the head of LINAC and the AbLS sample is 1 m, and the distance between the AbLS sample and the camera is 50 cm. The AbLS sample and camera were placed on a flat table. The digital camera was positioned at 90 degrees relative to the beam axis. A partition with tiny hole was placed in front of the camera to avoid background or stray light.

**Figure 2 sensors-22-04876-f002:**
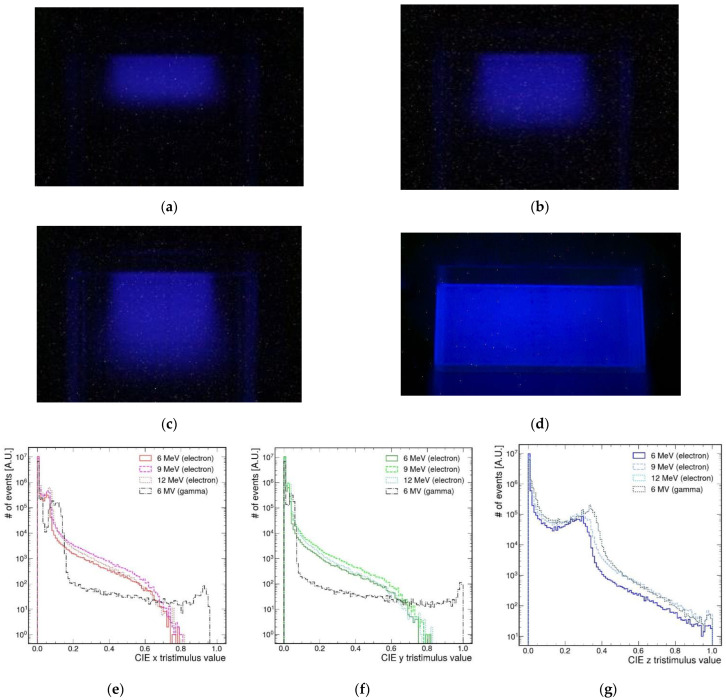
Subfigures (**a**–**c**) represent the fluorescent light of AbLS with 6, 9, and 12 MeV electron beams, and (**d**) represents the fluorescence of gamma rays from 6 MV electron. For electron beam or gamma rays, Varian Novalis Tx system was used. The beam was injected from top to bottom. The fluorescent light from AbLS is seen in the central part of the container. The glimmering light seen at the top and the bottom of the picture is due to the reflection from the container. (**e**–**g**) Decomposed histograms were based on CIE XYZ for each fluorescence image. RGB components were extracted from the photographed images in (**a**–**c**). Peaks around 0.3 in the z-value distribution were seen, compared to those of the x value or y value. Blue component was more dominant compared to the red and green components.

**Figure 3 sensors-22-04876-f003:**
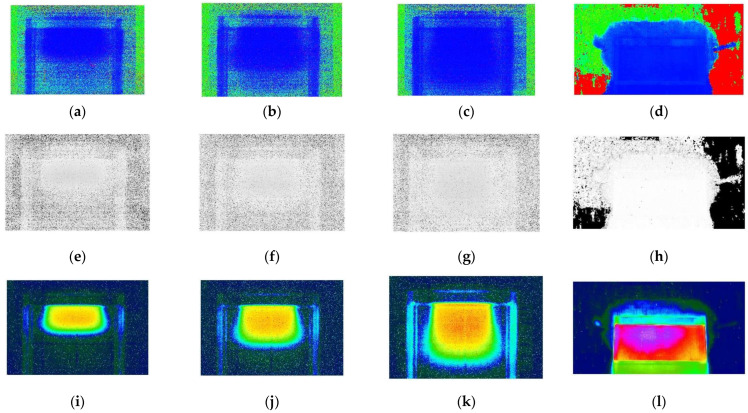
Reconstructed images of fluorescent light of 6, 9, and 12 MeV electron beams and gamma ray (last column (**d**,**h**,**l**)) in HSV space. (**a**–**d**) The blue component was dominant in the single H-based decomposition. (**e**–**h**) A single S basis represents achromaticity. Darker colors represent more achromatic colors, while lighter colors represent more colorful properties. (**i**–**l**) A single V decomposition displays luminance levels. For visibility, an arbitrary color system was used instead of a grayscale representation. The value of V increases from blue to magenta.

**Figure 4 sensors-22-04876-f004:**
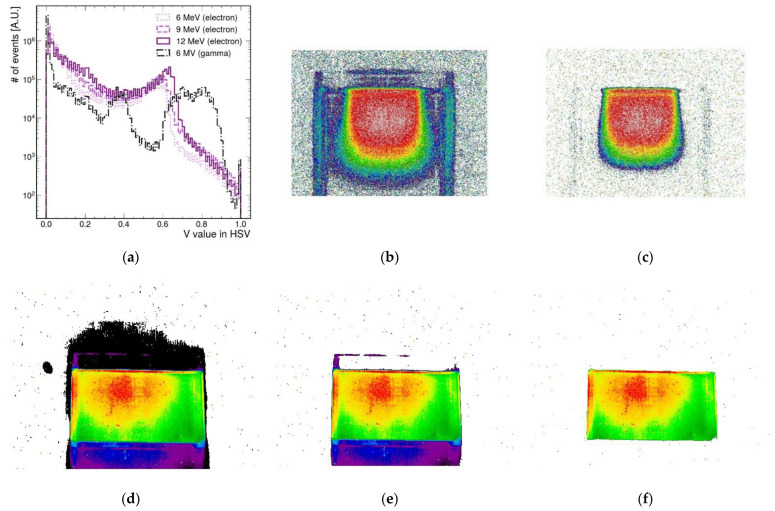
(**a**) A histogram of V values for each fluorescence image. Double peaks were seen with the gamma ray. One peak near 0.3 was due to reflected fluorescent light from the top and bottom of the container. Another peak near 0.6 was coming from the direct fluorescent light of AbLS. (**b**,**c**) Reconstructed images of a fluorescence image of a 12 MeV electron beam. (**d**–**f**) Reconstructed images of a fluorescence image of gamma ray. The differences in image reconstructions with V thresholds of 0.1, 0.3, 0.1, 0.3, and 0.6 were applied in (**b**–**f**). The regions where the AbLS sample received a lot of energy from the incident particle beam had high V values in the reconstructed image. For quantitative image pixel analysis, V-cuts were used to distinguish between signal and noise.

**Figure 5 sensors-22-04876-f005:**
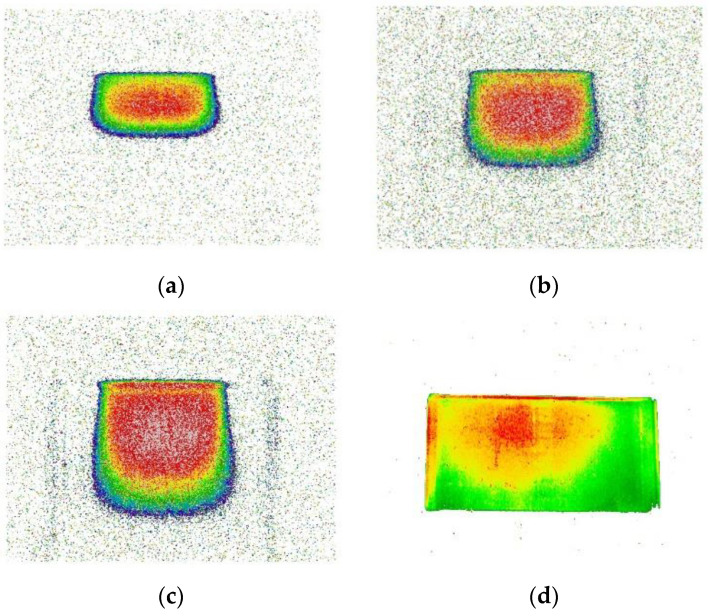
(**a**–**c**) Only pixels with a V value of 30% or more were selected among the 6, 9, and 12 MeV electron beam fluorescence images to reconstruct the photo image. The contour color indicates the amount of energy absorbed by AbLS. Red color means more energy absorption. (**d**) Only pixels with a V value of over 60% were selected in the gamma ray image. The 2D PDD MC simulation results with AbLS for 6, 9, 12 MeV electron beams and gamma ray are shown in (**e**–**h**), respectively. The solid blue line indicates the boundary of the container, the dashed blue line indicates the open top side of the container, and the white hatched area represents the area of the incident particle beam. More cyan, blue, and violet colors exist in the MC dose distribution. The dose in the MC and pixel image increases from blue to magenta.

**Figure 6 sensors-22-04876-f006:**
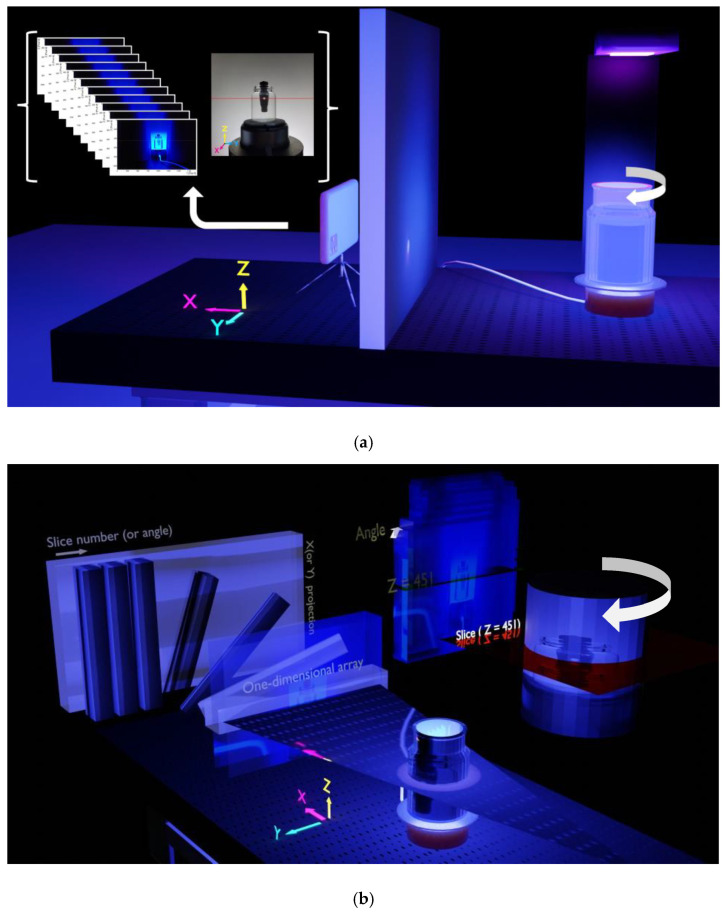
(**a**) A schematic of the sinogram with the combination of AbLS sample and the cell phone camera. A UV lamp was positioned approximately 10 cm above the AbLS sample with a T-shaped bar. The emission wavelength of the UV lamp was 365 nm. A 3 cm diameter 100 mL vial containing AbLS had a T-shaped bar with a microscopic hole in the center of the bar. The holes in the T-shaped bar immersed in the AbLS container can be seen. The AbLS sample vial was placed on an electric rotating disk. Then, the disk rotated at a constant speed. The rotated images were taken and extracted. The baseline for reconstructing the sinogram is indicated by the red solid line. The tomographic image was extracted and reconstructed based on this red line. This line corresponded to z = 451 pixel line value. (**b**) The specific z line in the photo corresponds to line data (or one-dimensional array) obtained from fixed angle. Then, by stacking these pixel line data from all angles, sinogram was formed. The sinogram was reconstructed from z = 451 pixel lines of each image.

**Figure 7 sensors-22-04876-f007:**
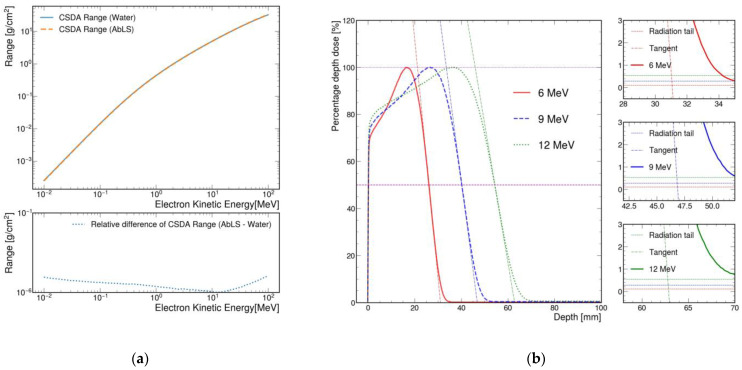
(**a**) (**Top**) The specific CSDA range curves were calculated from water and AbLS for 10^−2^–10^2^ MeV electron beams. The specific CSDA ranges of AbLS for 6, 9, and 12 MeV electron beams were 3.04, 4.50, and 5.89 g/cm^2^. When divided by the density (0.945 g/cm^3^) of AbLS, the CSDA ranges of AbLS were 3.22, 4.76, and 6.23 cm. (**Bottom**) The calculated specific CSDA ranges for AbLS and water were almost the same. (**b**) The projected ranges with MC simulations were 3.11, 4.68, and 6.28 cm. The projected ranges measured with the fluorescent image were 3.17 ± 0.30, 4.75 ± 0.30, and 6.41 ± 0.60 cm. The three insets on the right side were enlarged regions near the intersection of background and the extrapolated linear line for accurate practical range measurement.

**Figure 8 sensors-22-04876-f008:**
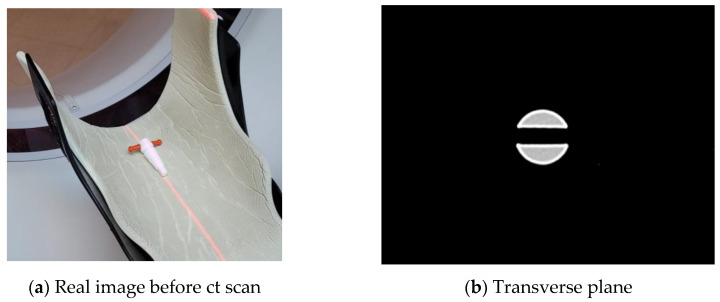
(**a**) The T-shaped bar placed on the CT examination table before scan. (**b**–**d**) The reconstructed CT images of the transverse plane, median plane, and frontal plane, respectively, for the T-shaped bar. Because CT has high resolution, it is possible to distinguish the difference in density, so even the detailed structure of the T-shaped bar can be seen.

**Figure 9 sensors-22-04876-f009:**
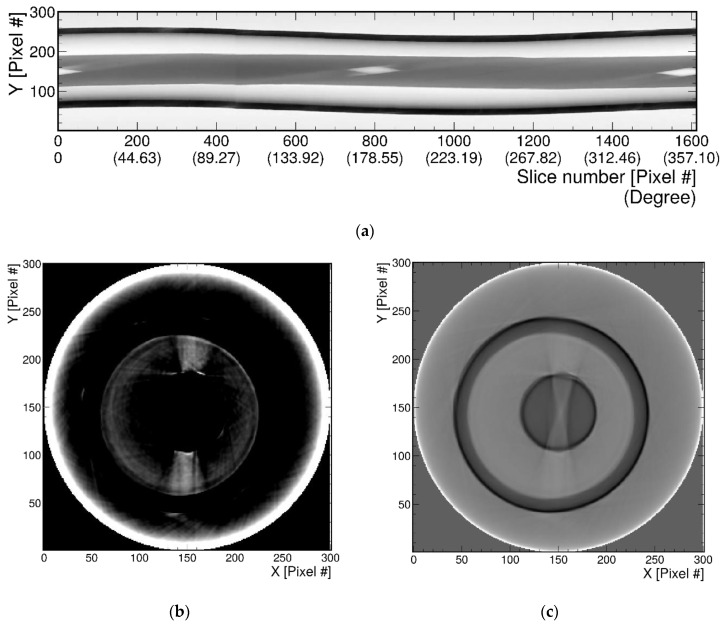
(**a**) 2D-pixel sinogram arrangement using the red baseline of the image inserted in Figure 6. (**b**,**c**) Reconstruction of the tomographic top-view images. White color in optical tomography represents the dark part in CT image, while black color represents the white part. (**b**) A narrow scale contrast was applied. The large outer circle shown here is the boundary of the glass bottle, while the small inner circle is the cross-sectional boundary of the T-bar. The two rectangles located in the coordinates (150, 100) and (150, 200) are cylinder-shaped holes across the center of the cross-section of the T-bar. Due to the narrow contrast range, rectangular region from (150, 110) to (150, 190) is the part that connects the upper rectangle and the lower rectangle, and it was not visible. (**c**) A wide-scale contrast was applied. The upper and lower rectangular connection, which was not seen in (**b**), can be seen. Unlike CT images, which used X-rays, the pin-wheel-shaped image at the center was generated by the optical effect of visible light. At the same time, a small circle appeared in the center of the rod, and it was created by ill-posed conditions, in which many solutions exist in inverse problems.

## Data Availability

Not applicable.

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
