# Peer review of "Pixel Image Analysis and Its Application with an Alcohol-Based Liquid Scintillator for Particle Therapy"

_sensors, 2022, doi:10.3390/s22134876_

Round 1

Reviewer 1 Report

I have read the manuscript with interest. Please see my comments in the attached file for details.

Authors' names, addresses, phone number should be reviewed and follow standard practice. 

Some abbreviations (CMOS, MC etc.) need to be corrected.

Figure caption for Fig. 4 was written over two pages. 

Figure 5 (d) and (h) are not displayed correctly.

In summary, two units needs to be superscripted.

It is good idea to get proofreading service from a native English speaker.

Author Response

Please take a look at an attached file. Thanks.

Reviewer 2 Report

This research is based on using a solution of alcohol as a liquid scintillation dosimeter for electron beams or in general for charged particles

The authors didn't show that this scintillating material is only suitable for charged particles. They proposed application for it in the field of electron beam radiotherapy which is giving way to special x-ray treatment techniques. However, such beams are always contaminated with x-rays. So will such x-rays also fluoresce this material? This has not been discussed.

The way the method is presented it seems that the images obtained are used to obtain slices not in CT mode but in CBCT mode which is not mentioned by the authors

Moreover, they didn't discuss the limitations of this imaging modality such as the scatter which influences the intensity measured by each pixel in the mobile phone camera

The method is not clear. For instance how are the slices obtained? and are the intensities extracted and formulated into sinograms? etc..

Also there is no comparison between range values obtained by this LS and the other ones and parallel plate ionisation chambers?

Authors mentioned that the concentration of this alcohol is small to the level that it does not affect the density it is still tissue equivalent. Yet they did not mention how much the conversion efficiency of this alcohol? is it high enough to detect even small doses? they need such calculations.  

Language: it is adequate but some sentences are ambiguous I highlighted some in the attached file but they need to review the whole paper.

Also, it seems that the article is written in a hurry for instance in one of the figures the caption states "6 MeV gamma rays from linac?" please check the whole paper for such mistakes before submitting it again

Author Response

To dear Reviewer #2

Please take a look at an attached file. Thanks.
